

# A Miocene breeding ground of an extinct baleen whale (Cetacea: Mysticeti)

Cheng-Hsiu Tsai

Department of Geology and Paleontology, National Museum of Nature and Science, Tsukuba, Japan

## ABSTRACT

Locating breeding sites is definitely a key to understanding the ecological requirements and maintaining the sustainability of populations/species. Here I re-examined published specimens of an extinct baleen whale, *Parietobalaena yamaokai*, from the lower part of Itahashi Formation (16.1–15.6 Ma, Middle Miocene) in Shobara, Hiroshima, Japan. A critical and previously unnoticed feature, the open suture between the supraoccipital and exoccipital, in one specimen indicates the preservation of a very young individual–under six months old and even close to a new-born calf. Given the occurrence of a new-born whale and relatively abundant assemblage of *Parietobalaena yamaokai*, I propose a previously hidden and unknown breeding ground for the extinct baleen whale, *P. yamaokai*, in the Middle Miocene of Shobara (16.1–15.6 Ma), Hiroshima. Discovery of paleo-breeding sites of extinct populations/species should further help us to understand biological extinctions from a long-term perspective as conservation paleobiology aims to offer new insights into policy making for conserving endangered populations/species.

## INTRODUCTION

Finding out the previously unknown breeding sites definitely is crucial for acquiring critical information for the conservation purpose of extant species. However, it is equally important to locate breeding grounds of extinct populations/species in the deep past (i.e., conservation paleobiology, see *Dietl et al., 2015*; *Barnosky et al., 2017*). This will provide critical information that can be used to develop more insightful conservation policies by integrating long-term perspectives that may be relevant to rapidly changing environments. For example, the eastern North Pacific population of gray whales, *Eschrichtius robustus*, is currently large and stable, and even becomes an attraction for whale watching. In fact, this population used to be close to the brink of extinction in the early 20th century, but the population recovered, thanks to the well-known breeding sites so that it facilitated the recovery and protection of a nearly extinct population. On the contrary, the status of the western North Pacific population is more uncertain, with their population down to 100 individuals, it could face the similar as their North Atlantic counterparts (*Mead & Mitchell, 1984*). As a result, discovery of a prehistoric breeding

Corresponding author
Cheng-Hsiu Tsai,
craniata@gmail.com,
cheng-hsiu.tsai@otago.ac.nz

site for the western North Pacific population (*Tsai et al., 2014*) should then bring a new consideration of how to further maintain and even recover the population of a near-extinct population.

Here, I report a Miocene breeding site for an extinct baleen whale, *Parietobalaena yamaokai* (Cetacea: Mysticeti) from Shobara, Hiroshima, Japan (16.1–15.6 Ma, Itahashi Formation, Bihoku Group). The finding of a previously unknown breeding ground of an extinct baleen whale should give us an opportunity for a new look at extinction event and further understand the ecological and evolutionary response of species to environmental changes from a long-term perspective as the conservation paleobiology aims to offer a new aspect for policy making when concerning the conservation of endangered populations/species.

## MATERIALS & METHODS

Published specimens of *Parietobalaena yamaokai* were re-examined, including HMN-F00023, HMN-F00024 (two accession numbers belong to the same individual, which was assigned as the type specimen), HMN-F00042, HMN-F00044, HMN-F00054, HMN-F00127 (all described in one paper; *Otsuka & Ota, 2008*), HMN-F00004 (*Kimura et al., 2010*), and HMN-F00640 (*Kimura et al., 2011*). All specimens were recovered from the same geological horizon–lower part of the Itahashi Formation, Bihoku Group; according to the occurrence of some nannofossils, such as *Sphenolithus heteromorphus* and *Helicosphaera ampliaperta*, this horizon corresponds to the NN4 biozone in the Middle Miocene (*Martini, 1971*; *Yamamoto, 1999*) and spans roughly from 16.1 to 15.6 Ma.

Determination of ontogenetic ages follows the results of studying early juvenile specimens and calves of extant *Eschrichtius robustus* (gray whale), *Balaenoptera acutorostrata* (minke whale), *B. physalus* (fin whale), and *Megaptera novaeangliae* (humpback whale) (*Walsh & Berta, 2011*); here, I included a fetal specimen of the blue whale, *Balaenoptera musculus* (USNM 268001), to show the open suture between the supraoccipital and exoccipital and facilitate the comparison with a very young individual of *Parietobalaena yamaokai* (Fig. 1). The phylogenetic placement of *Parietobalaena yamaokai* remains uncertain; for example, phylogenetically, *P. yamaokai* can be a balaenopteroid (*Marx & Fordyce, 2015*) or a "cetothere" *sensu lato* (*Boessenecker & Fordyce, 2017*). Given that the position of *P. yamaokai* is a balaenopteroid in the Marx and Fordyce or belongs to a sister lineage of balaenopteroids (a cetothere *sensu lato*) in the *Boessenecker & Fordyce (2017)*, the sequence of ossification and the ontogenetic age inferred from the fusion of occipital joints in other extant balaenopteroids (the gray whale, minke whale, fin whale, and humpback whale are all balaenopteroids) should still be a reliable proxy for *P. yamaokai*.

In addition, the body size could be an alternative approach to estimate the ontogenetic age, although the growth curve or full-grown size of *Parietobalaena yamaokai* remains unknown. I use two equations, both of which rely on the bizygomatic width of the skull to reconstruct the body size specifically for baleen whales (*Lambert et al., 2010*: $y$ (body length) $= 8.209^{\star}x$ (bizygomatic width) $+ 66.69$; *Pyenson & Sponberg, 2011*: log (body length) $= 0.92^{\star}$(log (bizygomatic width) $-1.64$) $+ 2.67$), to assess the total length of HMN-F00127.

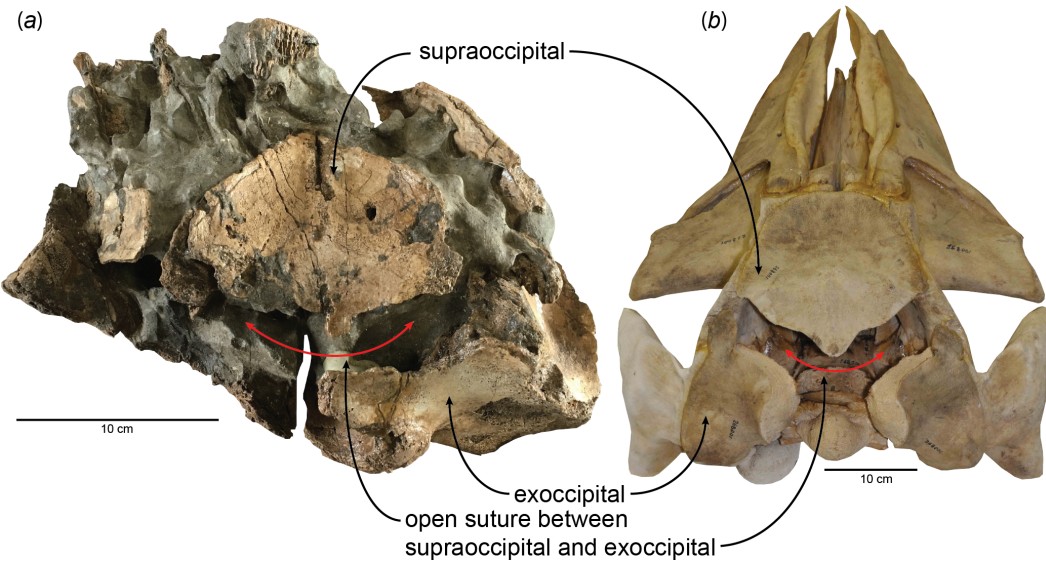

**Figure 1** Open suture between the supraoccipital and exoccipital in (A) a fossil baleen whale, HMN-F00127 (*Parietobalaena yamaokai*) and (B) a fetal specimen of blue whale, *Balaenoptera musculus* (USNM 268001).

Given the preserved condition, the exact bizygomatic width of HMN-F00127 is uncertain, but likely approximates 30 cm. In order to have a better understanding of their body size, a larger individual, HMN-F00042, whose bizygomatic width approaches but slightly less than 50 cm, was also estimated.

## RESULTS

New observations on the published specimens of *Parietobalaena yamaokai* show that several reported fossils are juvenile specimens, although it remains problematic for clearly identifying exact ontogenetic ages of each fossil specimen. However, one specimen (HMN-F00127) appears to be a very young individual, under six months old, judging from the open suture between the supraoccipital and exoccipital (Fig. 1) as the sequence and timing of the occipital ossification in the extant balaenopteroids provide an applicable proxy (*Walsh & Berta, 2011*). Given the preserved morphology of the supraoccipital in HMN-F00127, it is slightly eroded, but the overall edge remains intact, in turn, excluding the possibility of damage. Likewise, the separation between the supraoccipital and exoccipital of HMN-F00127 is unlikely to result from some unusual occipital fenestrations in some cetaceans (*Gao & Gaskin, 1996*; *Trimble & Praderi, 2008*), leading to the conclusion that the open suture is genuine, which could be comparable to a fetal/new-born blue whale, *Balaenoptera musculus* as shown in Fig. 1.

Additionally, given the estimated bizygomatic width (30 cm), the body size for HMN-F00127 was approximately 313 and 479 cm from *Lambert et al. (2010)* equation and *Pyenson & Sponberg (2011)* equation, respectively. Similarly, a larger specimen of *P. yamaokai*, HMN-F00042, whose bizygomatic width is slightly worn down, but approaches 50 cm, shows a physically immature feature—unfused vertebral discs. The unfused vertebral discs

can be a referable feature to estimate the ontogenetic age (*Moran et al., 2015*), but can only be a rough proxy. With 50 cm, the body size of HMN-F00042 ranges from 477 cm (*Lambert et al., 2010* equation) to 766 cm (*Pyenson & Sponberg, 2011* equation).

## DISCUSSION

HMN-F00127 represents a rare baleen whale fossil whose ontogenetic age was previously unrecognized and actually can be genuinely determined the ontogenetic age to some extent–under six months old, given the open suture between the supraoccipital and exoccipital. Furthermore, the supraoccipital is virtually floating, lacking a firm contact with surrounding bones (e.g., exoccipital and parietal) and in turn suggesting a much younger developmental stage (likely to be close to a new-born calf?), comparable to a fetal specimen of blue whale, *Balaenoptera musculus* (Fig. 1). In addition, a relatively abundant assemblage of *Parietobalaena yamaokai* (*Otsuka & Ota, 2008*; *Kimura et al., 2010*; *Kimura et al., 2011*) from the same geological horizon (Itahashi Formation, Bihoku Group) and geographical area (Shobara, Hiroshima) invites a sensible interpretation: the presence of a previously unknown Miocene breeding/calving site for baleen whales in Shobara, Hiroshima. Likewise, the reconstruction of paleogeography and paleoclimatology of the area where HMN-F00127 and other specimens of *P. yamaokai* were found (Fig. 2; also see *Noda & Goto, 2004*) suggests a relatively protected and possibly an ideal locality for breeding purposes as extant baleen whales seek for calving sites (*Hindell, 2009*; *Rayment, Dawson & Webster, 2015*).

However, some doubt may come from the species identification as juvenile morphology likely differs from adults substantially in extant baleen whales. The underlying heterochronic process that affects the evolutionary history and developmental pattern of *Parietobalaena yamaokai* remains unidentified: clades showing paedomorphic neoteny have similar morphology, which allows reliable identification based on juvenile specimens (e.g., the pygmy right whale, *Caperea marginata*; *Tsai & Fordyce, 2014a*; *Tsai & Fordyce, 2014b*). By contrast, peramorphic acceleration clades show disparate morphologies between juveniles and adults, hence likely resulting in misleading interpretations (e.g., the humpback whale, *Megaptera novaeangliae*; *Tsai & Fordyce, 2014a*; *Tsai & Fordyce, 2014b*; or, an extinct eomysticetid, *Waharoa ruwhenua*; *Boessenecker & Fordyce, 2015*), making the conspecific confirmation of fossil specimens between juveniles and adults problematic. In addition, several baleen whale species apart from *P. yamaokai* were found from the same locality, such as *Diorocetus shobarensis* and *Hibacetus hirosei*, complicating the scenario of species identification of juvenile specimens. As juvenile identification of *Parietobalaena yamaokai* and/or other contemporaneous baleen whales in Shobara, Hiroshima is beyond the scope of this study, here I follow the interpretations of previous studies on HMN-F00127 as a *P. yamaokai* (*Otsuka & Ota, 2008*; *Kimura et al., 2010*; *Kimura et al., 2011*).

Additionally, it is worth noting that if the estimated body size for HMN-F00127, 313 cm from *Lambert et al. (2010)* equation or 479 cm from *Pyenson & Sponberg (2011)* equation, is approximately accurate for a young individual under 6 months old, *Parietobalaena yamaokai* would have been a middle-sized baleen whale, 12–15 m when full-grown, similar

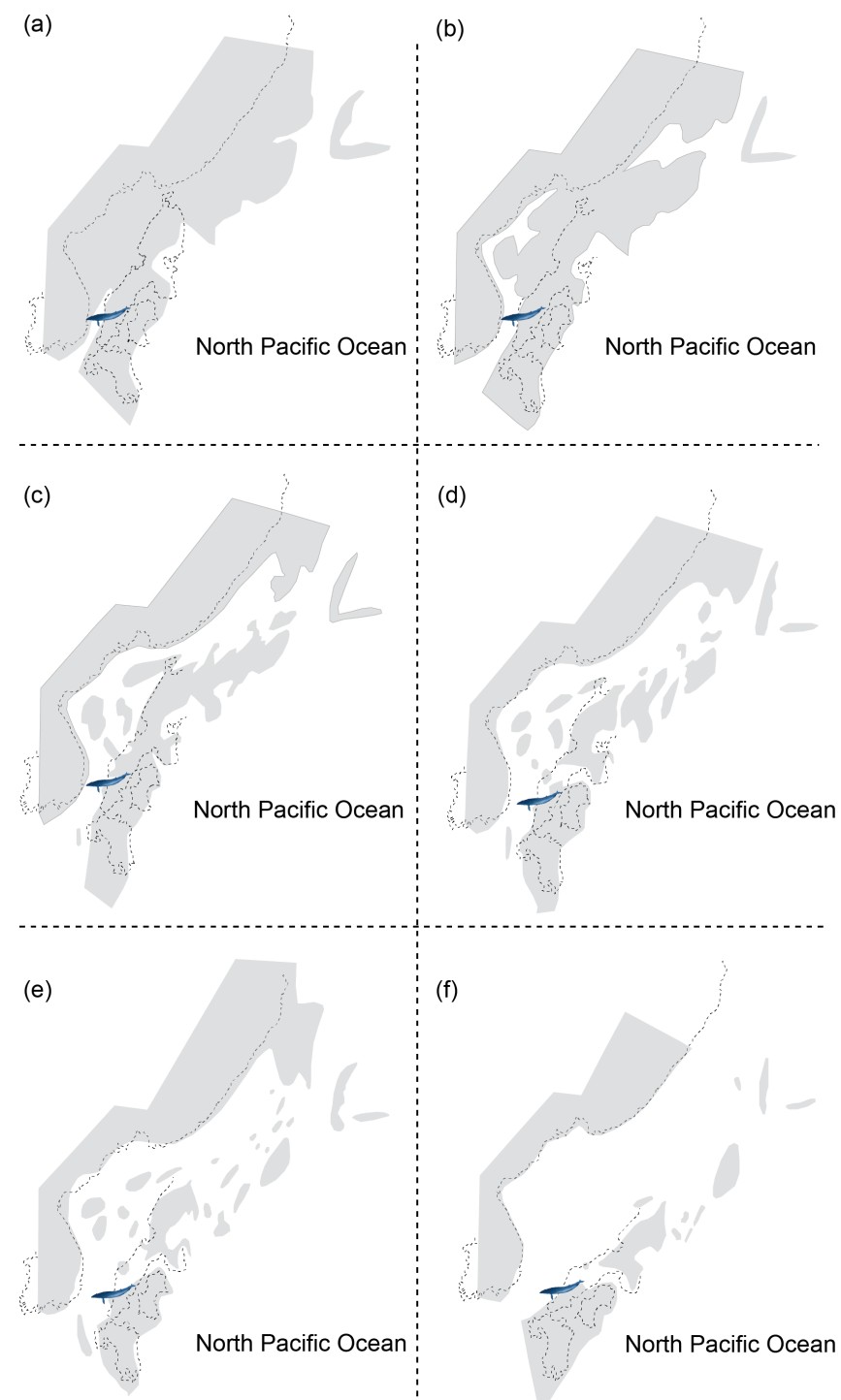

**Figure 2** Reconstructing maps of Japan and surrounding areas during the Middle Miocene, particularly ranging from 17 to 15 Ma as this period corresponds to the geological age of HMN specimens and then indicates the change of geography over time: (A) 17 Ma; (B) 16.75 Ma; (C) 16.5 Ma; (D) 16.25 Ma; (E) 16 Ma; (F) 15 Ma. Modified from *Noda & Goto (2004)*.

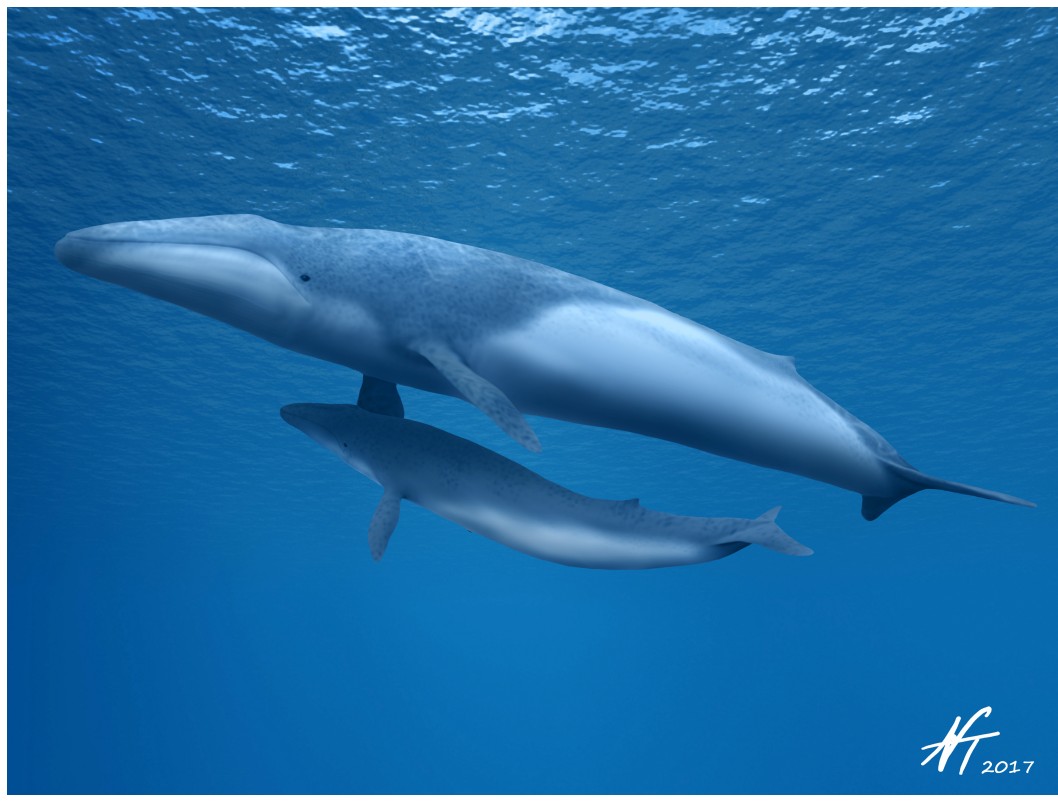

**Figure 3** **Life restoration of a mother-calf pair of *Parietobalaena yamaokai*** (©Nobumichi Tamura).

to the size of extant gray whales, *Eschrichtius robustus* (see *Andrews, 1914*; *Rice & Wolman, 1971* for references to the size of gray whales). This can be corroborated by a larger, but physically immature specimen HMN-F00042, whose estimated body length is 477 cm or 766 cm under different equations. HMN-F00042 may also be a juvenile, ontogenetically older than HMN-F000127, given the presence of several loose and unfused vertebral discs, but unable to further pin down its precise age at present. As a result, if the interpretation on the estimated size and ontogenetic age for HMN-F00127 or HMN-F00042 is correct, it then suggests the existence of large baleen whales (over 10 m) in the Middle Miocene (Itahashi Formation, 16.1–15.6 Ma), substantiating the early origin of baleen whale gigantism (*Tsai & Kohno, 2016*), instead of a recent origin (*Slater, Goldbogen & Pyenson, 2017*), albeit the lack of ancestor-descendant relationships to illustrate the detailed evolution of gigantism for now (see *Tsai & Fordyce, 2015* for discussion of ancestor-descendant relationships).

Regardless, identifying a possible Miocene breeding site for baleen whales in the northern hemisphere also raises some interesting questions: when, where, and which species of baleen whales initiated the annual, long migration between feeding and calving grounds? If the interpretation in this paper is correct, it then represents the earliest known site (Middle Miocene, 16.1–15.6 Ma) for baleen whale breeding in the northern hemisphere (Fig. 3). However, given the presence of a young individual of an Oligocene baleen whale, *Waharoa ruwhenua* (Cetacea: Eomysticetidae), together with isotopic analyses

of other eomysticetid species (for example *Tokarahia*, *Clementz et al., 2014*), *Waharoa* or other eomysticetids may have established a long, latitudinal migration and come to continental waters of the Oligocene of New Zealand for breeding in the southern hemisphere (27.3–25.2 Ma, *Boessenecker & Fordyce, 2015*). In view of extant baleen whales, annual migration between feeding and breeding sites may not result from long-term evolutionary consequences, but instead ecological factors. In other words, establishment of having regular migrations in baleen whales may occur multiple times. On the other hand, there are several hypotheses, interpreting why baleen whales have long-distance migrations annually, such as thermoregulatory purpose for calves or reducing risk of predation by killer whales, *Orcinus orca* (*Corkeron & Connor, 1999*). Accordingly, seeking for suitable breeding and feeding sites and then resulting in annual migrations in baleen whales may have evolved and appeared independently in different lineages, by different ecological causes, and at different geological timings. More fossils and higher resolution of paleo-geographical and paleo-climatic reconstructions should further test hypotheses presented in this paper and explore more details of behavioral evolution in baleen whales, whose behavioral evolution may shape the direction of evolution in the marine regime as baleen whales are the largest vertebrates in the history of Life.

### Institutional Abbreviations

**HMN**    The Hiwa Museum for Natural History, Shobara, Hiroshima, Japan
**USNM**    National Museum of Natural History, Smithsonian Institution, USA

## ACKNOWLEDGEMENTS

I thank Tomomi Kiyoshi, Hitoshi Ohzawa, Yoshio Furukawa, Takato Ueda for inviting and encouraging to study HMN specimens; James Mead, Charles Potter, John Ososky, Nicholas Pyenson, and David Bohaska for collection access and allowing photography (USNM specimens); Nobumichi Tamura for illustrating the life restoration (Fig. 3); the handling editor John Hutchinson for helpful guidance; Brian Beatty and one anonymous reviewer for constructive comments and suggestions; Megumi Saito-Kato and Atsushi Yabe for discussing paleogeography of Japan; James Mead (Washington, D.C.) for accommodating at the "Happiness Hotel" during various research visits.

### Funding

CHT was supported by a Japan Society for the Promotion of Science Postdoctoral Research Fellowship. The funders had no role in study design, data collection and analysis, decision to publish, or preparation of the manuscript.

### Grant Disclosures

The following grant information was disclosed by the author:
Japan Society for the Promotion of Science Postdoctoral Research Fellowship.

## Competing Interests

The author declares there are no competing interests.

## Author Contributions

- Cheng-Hsiu Tsai conceived and designed the experiments, performed the experiments, analyzed the data, contributed reagents/materials/analysis tools, wrote the paper, prepared figures and/or tables, reviewed drafts of the paper.

## Data Availability

Specimens are at the Hiwa Museum for Natural History, Shobara, Hiroshima, Japan, and the National Museum of Natural History, Smithsonian Institution, USA.

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
