# Peer review of "A Miocene breeding ground of an extinct baleen whale (Cetacea: Mysticeti)"

_PeerJ, doi:10.7717/peerj.3711_

## Round 0.1 · original submission · Major Revisions

Apologies for the delay but we have received two constructive reviews now and they agree that the paper needs improvements. It must not over-sell the evidence and needs to consider whether alternative hypotheses are truly being tested. Please take care with this and address all reviewers' points individually in a Response document. We will need to have the paper re-reviewed to see if the reviewers are more convinced. Thank you for submitting this manuscript.

·

Basic reporting

No additional comments on this topic. The paper is sound in this regard.

Experimental design

The conclusion that is presented regarding the locality being a calving ground isn't explicitly tested as a hypothesis with any alternative hypotheses. In fact, no clear hypothesis is stated, and it would be better to make clear separate the findings from the speculation. Two main points: 1) I am concerned that the evidence for what is tested (the specimen being a juvenile) could use more rigor, or alternative explanations could be stated. 2) Likewise, the speculations derived from the results could be more explicitly stated as speculations, and alternative explanations for those results should also be considered and included.

1) The real question being explored is whether the specimen is a juvenile fossil mysticete. The author presented some evidence of this, including body mass estimates and macroscopic views of a skull that appears to have a similar open suture pattern as other young modern balaenopteroids. I agree that this is a compelling similarity, and I think the authors are likely to be interpreting this correctly. However, we all have seen the thin bone of the occipital regions of cetaceans crack and have holes in them, especially in those regions, and it would be better if there were some close up images (perhaps microscopic) that would somehow demonstrate that these aren't simply broken edges. I understand that it may be hard to do, but in that case it is important to be clear about limitations of the specimen.

2) The presence of a juvenile this young need not indicate a calving ground in the same way that many (not all) mysticetes today have locations for calving. The distinction is actually summarized very well in the paper they cite about why whales migrate, and I would urge the author to more carefully explore the alternative hypotheses of behaviors that are explored there. Also, it is important to keep in mind the fauna, paleoclimate and oceanic patterns of the Miocene that were different than today. The conclusions of that article are mostly indicative of the role of orcas in driving predator avoidance behaviors in some modern balaenopterids, partly because of the preferred prey of pinnipeds. This paper would be improved by exploring the alternative ideas about what this juvenile specimen could indicate. Did these taxa calve there to avoid some other predator? How could other primary prey species like pinnipeds been part of that equation? Did these mysticetes even have to migrate? Some of these would be better tested by exploring whether some sites have a disproportionate number of juveniles than adults (and you'd need good sample sizes of multiple populations)? Good papers to consider for insights on how demographics of fossil populations could be used in this way are typically on terrestrial animals (Mihlbachler, 2003 Paleobiology "Demography of late Miocene rhinoceroses (Teleoceras proterum and Aphelops malacorhinus) from Florida: linking mortality and sociality in fossil assemblages"), though life tables for whales exist (Albert Myrick papers in the International Whaling Commission reports comes to mind).

Validity of the findings

Same as for the comments on experimental design.

Additional comments

This is an important piece of information on the distribution of early ontogenetic stages of cetaceans, and the author should be thanked for putting the effort to make sure this gets in the literature. I would like the alternative hypotheses to be more explicitly stated here, and even more importantly, to see more thorough assessments of the ontogenetic stages and demographics of populations of fossil cetaceans more thoroughly reported like this and then synthesized.

Reviewer 2 ·

Basic reporting

See general comments.

Experimental design

See general comments.

Validity of the findings

See general comments.

Additional comments

This manuscript is very thought provoking and original work. This work should be of interest to other cetacean paleontologist, neontologist and has the potential to be relevant to researchers interested in cetacean conservation and life habits.

The manuscript is generally well written, although I found some sentences to be overly long (see detailed comments). The figure is of high quality and informative. However, I do have some concerns that are outlined in detail below and that I hope the author take into consideration.

General comments:

My main concern with this manuscript is that I am having a hard time seeing how this ancient nursery site could be relevant to breeding grounds of modern taxa, conservation and the effects of changing environments. I suggest the author broadens the discussion to include possible factors that may have affected these ancient populations as a possible connection to the statement about conservation paleobiology. For example, was this population affected by global changes after the middle Miocene Climatic Optimum? If this cannot be done, then I suggest the author limits the paper to reporting this ancient nursery site, which is interesting on its own. I understand the importance to bring to attention the highly reduced population of western North Pacific gray whales, but maybe this is not the best connection, specially since the taxon used in this work is not even part of crown Balenopteroidea. In addition to this, I have some specific comments detailed below.

Specific comments:

Lines 28-32: This sentence is a bit too long, I suggest shortening it or splitting it into two. For example, maybe something like this: “However, it is important to locate breeding grounds of extinct populations/species in the deep past (i.e. conservation paleobiology: Dietl et al., 2015; Barnosky et al., 2017). This will provide critical information that can be used to develop more insightful conservation policies by integrating long-term perspectives that may be relevant to rapidly changing environments.”

Lines 32-37: This sentence is a bit too long and could be split.

Lines 37-40: This sentence is not entirely clear, I suggest changing it to “On the contrary the status of the western North Pacific population is more uncertain, with their population down to 100 individuals, it could face the similar fate as their North Atlantic counterparts (Mead and Mitchell, 1984).”

Line 45: I suggest changing “…a hidden and lost breeding ground…” to “… a previously unknown breeding ground…” saying “lost” implies that it was once known, but then forgotten.

Line 86, Results: The author should add the number of individuals examined, and how many of them are juveniles and adults, even if its only a general interpretation of their ontogenetic age.

Lines 98-99: I suggest this part “HMN-F00127 represents a rare baleen whale fossil that previously was unrecognized and can be genuinely, to a certain extent, determined the ontogenetic age-under six months…” can be simplified, for example “HMN-F00127 represents a baleen whale fossil whose ontogenetic age can be determined to be under six months…”

Line 110: change “… protective and…” to “… protected and…”

Lines 114-123: This sentence is extremely long and the message gets lost, I suggest the author shortens it or splits it.

Lines 116-117: For clarity I suggest changing “… paedomorphic neonteny clades have similar morphology for reliable identification…” to “… clades showing paedomorphic neonteny have similar morphology which allow for reliable identification…”

Lines 130-141: This interpretation is a bit problematic. Extrapolating adult size for Parietobalaena yamaokai based on the estimated size of a juvenile may not be entirely accurate and lead to incorrect interpretations. If additional specimens are available, or have been published, then I strongly suggest the author uses whichever is the largest specimen available for an adult or advanced juvenile body size estimate using the formulas from Lambert et al. (2010) and Pyenson & Sponberg (2011).

---

## Round 0.2 · accepted · Accept

One reviewer is now satisfied and the other has not replied so I am OK with accepting this MS- congratulations!

·

Basic reporting

All is satisfactory

Experimental design

Thank you for your thoughtful responses. I understand the limitations you face with the project and consider your added text to address my comments as satisfactory.

Validity of the findings

Thank you for your thoughtful responses. I understand the limitations you face with the project and consider your added text to address my comments as satisfactory.

Additional comments

Thank you for your thoughtful responses. I understand the limitations you face with the project and consider your added text to address my comments as satisfactory.